# Analysis of the Behavior of Mass Concrete with the Addition of Carbon Nanofibers (CNFs) When Exposed to Fire

**Rubén Serrano Somolinos, María Isabel Prieto Barrio**[ID]**, María de las Nieves González García** *[ID] **and Kenzo Jorge Hosokawa Menéndez**

Escuela Técnica Superior de Edificación, Universidad Politécnica de Madrid, Madrid 28040, Spain;
ruben.serrano.somolinos@alumnos.upm.es (R.S.S.); mariaisabel.prieto@upm.es (M.I.P.B.);
k.hosokawa@upm.es (K.J.H.M.)
* Correspondence: mariadelasnieves.gonzalez@upm.es

**Abstract:** Due to the importance of concrete as a structural material and the pathologies that can be achieved by reinforced concrete structures when they are subjected to the action of fire both at the level of resistance and deformation, in this research we study the mechanical behavior of mass concrete with the addition of carbon nanofibers (CNFs) when exposed to the action of fire, in order to determine the improvements that this type of addition produces in concrete. To achieve this objective, compression break tests have been carried out on cylindrical concrete specimens incorporating CNFs. From the analysis of results, it can be concluded that the residual resistant capacity of concrete with the addition of 1% of CNFs by weight of cement subjected to the direct action of fire, is greater than that of concrete without additions, not obtaining better results, if the addition of CNFs increases to 2%. The addition of 1% of CNFs has not influenced the temperatures reached in the concrete, but produces a more homogeneous cooling and that the paste-aggregate bond is maintained despite thermal aggression, which decreases the spalling effect.

**Keywords:** carbon nanofiber; CNFs; compressive strength; post-cracking; spalling

## 1. Introduction

Concrete is the most important structural material in construction, from which durability, stability, resistance and a good response when exposed to fire are expected, preventing the appearance of deformations and loss of rigidity [1–3]. The incorporation of fibers in concrete modifies its behavior, principally in tension, preventing the opening and spreading of cracks, and increasing its ductility [4,5].

When concrete is affected by a fire, it experiences a loss of compressive strength in the order of 15% when the said fire reaches temperatures of 400 °C, whilst its mechanical capacity marginally reduces in temperatures of up to 280 °C [6]. Furthermore, its longitudinal modulus of elasticity decreases by 75% when temperatures close to 400 °C are reached, provoking a high number of cracks [7–11]. Numerous research studies in this field have proven that the incorporation of metallic or polypropylene fibers results in increased resistance and a reduction in cracking when the concrete is exposed to high temperatures [12–17].

On the other hand, progress in the field of nanomaterials is an inestimable opportunity to develop new materials, existing among others, nanofibers (CNFs), carbon nanotubes (CNTs) and graphene oxide (GO), which offer a further dimension for interacting with cement and with concrete. The effect of incorporating low dosages of nanomaterials delivers improvements in the hydration, microstructure and mechanical properties of cement-based materials, due to their electrical and

chemical properties [18,19]. The main disadvantages of the nanoscale use of these new materials are the lack of binding between the nanoparticles and the cement mixture and the high cost of producing them, although there are more economical options, such as carbon nanoplatelets, which improve watertightness and prevent chloride intrusion [20].

Studies carried out using graphene oxide (GO) show that the material possesses good chemical stability and high heat and electrical conductivity, making it ideal for manufacturing and use in different applications, particularly when good thermal management is required. Its use enables improvements to be achieved in concrete properties, increasing the tensile and flexural strength of the associated cement compounds and the transport phenomena which improve their durability. These phenomena are due to the fact that the incorporation of graphene oxide provokes hydration by adsorption of both water molecules and cement components, which improves the material's performance [21–27].

Meanwhile, it has been proven that improvements in the mechanical behavior of strain-hardening cementitious composites reinforced with graphene oxide are due to the increased resistance of the matrix, as well as the chemical bond between polyvinyl alcohol and the cement matrix due to the addition of graphene oxide, the amounts added being very important, since the addition of too much graphene oxide would have a detrimental effect [28].

Where carbon nanotubes (CNTs) are concerned, it has been observed that an adequate dispersion of carbon nanotubes increases compressive strength by up to 40% and moderately increases tensile strength but with a high level of deformations, taking into account that low concentrations of long CNTs are more effective than a high volume of short CNTs [29–32]. When it comes to the use of carbon nanotubes (CNTs) in concrete structures which are exposed to high temperatures, there is barely any research. Their use in the recovery of flexural strength in heat-damaged of reinforced concrete (RC) beams repaired using carbon fiber reinforced polymer compounds (CFRP) is recognized, it having been demonstrated that the addition of CNTs increased the capacity of the compound materials to recover the bending load capacity and rigidity of heat-damaged beams [33].

With regard to the behavior of carbon nanofibers (CNFs), there are numerous research projects in which they are added to polymer compounds, but not like this as an addition to cement. Research carried out in this field shows that the addition of CNFs to cement paste improves flexural strength, ductility, fracture toughness and micro-crack control, due to its large specific surface area, its surface chemistry and the narrow spacing of the carbon nanofibers [34,35]. Furthermore, the use of low CNF and CNT concentrations increases compressive strength at an early age, exhibiting greater dependence on the amount of fibers than on their degree of dispersion. On the other hand, the addition of CNTs at 0.1% of the weight and CNFs reduces the corrosion rate and significantly increases its strength, delaying the start of corrosion [36–38].

There are also research projects which demonstrate the sensitivity of these CNF compounds in detecting their own damage, in other words, the possibility of manufacturing structural damage sensors using CNFCCs is being studied, since the specimens with the addition of CNFs display a good capacity for detecting deformations for 28-day curing periods [39].

As has been seen in the bibliography consulted, the behavior of concrete and cement mortars when exposed to high temperatures has been significantly researched in the case where what is incorporated is steel or polypropylene fibers, but there are very few research projects which look at the behavior of cement compounds with the addition of nanomaterials which are exposed to high temperatures and as such, the objective of our work is to study the mechanical behavior of structural mass concrete with the addition of carbon nanofibers (CNFs) when its surface is directly exposed to fire, comparing the behavior before and after such exposure and with concrete without additions.

## 2. Materials and Methods

During the process and execution of the experimental part of the research, the materials and resources detailed below were used:

Materials used:

- Cement type CEM II/B-L 32.5 according to the Structural Concrete Code UNE-EN 197-1:2011 and RC-08 [3,40]. Its physical and chemical characteristics are shown in Table 1.
- Washed, siliceous, fine river sand with a particle size of 0–4 mm, according to regulation UNE-EN 13139/AC: 2004 [41].
- Washed, siliceous, coarse aggregate with a particle size of 4–20 mm, with a maximum aggregate size of 12 mm, according to regulation UNE-EN 12620:2003+A1:2009 [42].
- Water from the Canal de Isabel II supply system in the Madrid region, since this meets the technical specifications established for structural concrete.
- Carbon nanofibers (CNFs), used with approximate diameters of 100 nm, density of 1.95 $g/cm^3$ and a surface area of 45 $m^2/g$ with a separation between the graphite planes of 0.335–0.342 nm.
- No additives were used.
- The dosages which were used are shown in Table 2. The content of cement, sand, gravel and water is that used in each mix. The amount of addition used in each case is expressed in two ways: according to the percentage (%) of the cement weight and in $kg/m^3$ of concrete.

**Table 1.** Chemical composition and physical characteristics of the cement, sand, gravel and carbon nanofibers (CNFs).

| | | | | |
|---|---|---|---|---|
| Cement | $SO_3$ | ≤3.50% | Compression strength at 28 days | ≥32.5 MPa ≤52.5 MPa |
| | $Cl^-$ | ≤0.10% | Expansion (Le Chatelier) | ≤10 mm |
| | Chrome (VI) water soluble | ≤2 ppm | Setting start time | ≥75 min |
| | Compression strength at 7 days | ≥16 MPa | Setting final time | 220 min |
| Sand | $Cl^-$ | ≤0.005% | Sand equivalent | ≤75 |
| | Lightweight particle | ≤0.50% | Saturated density dry surface | 2.549 $g/cm^3$ |
| | Acid-soluble sulphate | ≤0.80% | Organic material | Exempt |
| | Total sulphur compounds | ≤0.11% | Water absorption | 3% |
| Gravels | $Cl^-$ | ≤0.001% | Saturated density dry surface | 2.711 $g/cm^3$ |
| | Acid-soluble sulphate | Category $AS_{0.2}$ | Lightweight organic contaminants | ≤0.50% |
| | Total sulphur compounds | ≤0.02% | Water absorption | ≤0.50% |
| CNFs | Density | 1.95 $g/cm^3$ | Diameter | 100 nm |
| | Surface area | 45 $m^2/g$ | Separation between the graphite planes | 0.335–0.342 nm |

**Table 2.** Dosage for the mixes and material content depending on the addition.

| Concrete Type | | HM-25 | |
|---|---|---|---|
| **Addition Type** | **Without Addition** | **CNFs** | **CNFs** |
| Nº specimens | 6 | 6 | 6 |
| Dosages(c/s/g/w) | | 1/2/3/0.5 | |
| Cement (kg) | 4.088 | 4.088 | 4.088 |
| Sand (kg) | 8.472 | 8.472 | 8.472 |
| Gravel (kg) | 14.616 | 14.616 | 14.616 |
| Water (l) | 2.040 | 2.040 | 2.040 |
| Addition (Kg/m$^3$) [1] | 0 | 2.3 | 4.6 |

[1] The addition of CNFs represents percentages of 1% and 2% of the weight of the cement, which is equivalent to 2.3 $kg/m^3$ and 4.6 $kg/m^3$ of concrete respectively.

With the aim of achieving the objective of the present research project, the behavior of mass concrete specimens without additions and with different CNF percentages has been studied in compressive strength tests in accordance with Annex 14 of the Structural Concrete Code [3] for the manufacturing of mass concrete reinforced with fibers. A group of test pieces was initially tested in direct contact with fire, before subsequently undergoing compression testing to breaking point, with the aim of comparing

its behavior with test pieces that were not exposed to thermal aggression, and establishing results relating to tensile strength, strain and maximum and ultimate strain energy densities.

In order to carry out the analysis, 18 cylindrical test pieces were made, measuring 100 mm in diameter and 200 mm in height, in line with UNE-EN 12390-1:2013 [43]. The testing program was organized using a single mix for the purpose of having the concrete manufactured at the same time and with the same characteristics.

Six mass concrete test pieces were made without the addition of CNFs, 6 test pieces were made with the addition of 1% CNFs to the cement weight and 6 test pieces were made with the addition of 2% CNFs. The addition of CNFs represents percentages of 1% and 2% of the weight of the cement, which is equivalent to 2.3 kg/m$^3$ and 4.6 kg/m$^3$ of concrete respectively. For each type of test piece, 50% of them directly underwent compression testing to breaking point, according to regulation UNE-EN 12390-3:2009 [44], whilst the remaining 50% previously underwent direct fire testing in line with the fire resistance tests for materials used in the Fire Prevention and Extinction Service of the Madrid region and with regulations UNE-EN 1363-1:2012, UNE-EN 1363-2:2000 and UNE-EN 1365-4:2000 [45–47], and once they had cooled down, they also underwent compression testing to breaking point.

The nomenclature and tests performed on the test pieces can be seen in Table 3.

**Table 3.** Nomenclature and tests performed on the mass concrete test pieces.

| Test | Nomenclature | | |
|---|---|---|---|
| Compressive strength testing | SA-P1 | GF-1%.P1 | GF-2%.P1 |
| | SA-P2 | GF-1%.P2 | GF-2%.P2 |
| | SA-P3 | FA-1%.P3 | GF-2%.P3 |
| Fire + compressive strength testing | SA-P4.F | GF-1%.P4.F | GF-2%.P4.F |
| | SA-P5.F | GF-1%.P5.F | GF-2%.P5.F |
| | SA-P6.F | GF-1%.P6.F | GF-2%.P6.F |

Firstly, and before starting to manufacture the concrete, the materials required to make the test pieces indicated in Table 1 were maintained in laboratory conditions for 24 h. The gravel was sifted using a sifting machine, with the aim of guaranteeing that there were no aggregates which were larger than 12 mm. Then, each of the materials to be used was weighed separately: cement, sand, gravel, addition of nanofibers and water, using industrial dial scales or digital scales, depending on the accuracy required.

The initial concrete mixing was performed by combining the dry materials by hand, adding the materials in the following order: gravel, cement, sand and finally the addition of CNFs, in order to obtain a perfectly homogeneous mixture and greater dispersion of the nanofibers. The dry mixture was subsequently transferred to the IBERTEST vertical shaft planetary mixer, model CIB-701 updated to IB32-040V0 (Ibertest Advanced testing solutions).

The mixer combined the dry materials once again for 2 min using its rotary blades, before gradually adding the water, taking into account the dampness of the aggregates used, until a plastic concrete consistency was achieved after 5 min of mixing.

Once the concrete was in a plastic, cool state, the test piece molds were filled in accordance with the fresh concrete testing regulations [48]. The molds were filled in sections, compressing each layer of concrete using a steel tamping rod to reach every part of the mold, before filling it up. Once the test pieces had been made, they were stored at a laboratory temperature of approximately 22 °C ± 3 °C and at an approximate relative humidity of 60% for 24 h, before being removed from the molds.

After 24 h, the test pieces were demolded and cured in a humidity chamber at a temperature of 20 °C ± 2 °C and a relative humidity of ≥95% for 28 days. During this time, the test pieces set and harden until they reach a minimum level of resistance that enables the compression to breaking point and fire tests to be performed, which will be detailed in the following section. The process of producing the mass concrete test pieces can be seen in Figure 1.

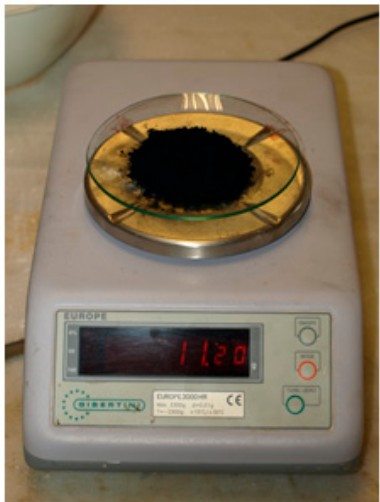
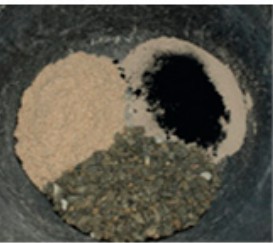
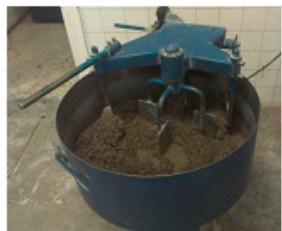
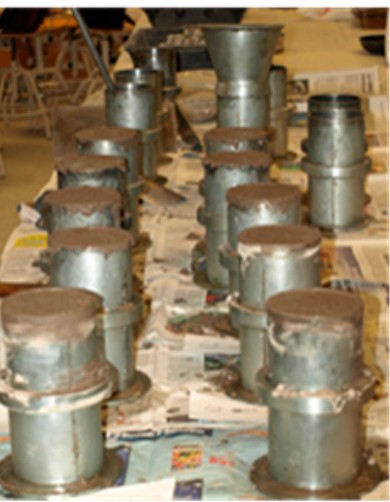

**Figure 1.** Process of manufacturing cylindrical test pieces with and without addition: dosage, standardization, mixing and molding.

In order to study the behavior of mass concrete, with and without additions, when exposed to fire, two types of test have been performed on the test pieces: a test exposing them to the direct influence of a real fire, carried out at the Madrid Fire Department, and a subsequent test involving compression to breaking point, which will enable us to compare the mechanical capacity of mass concrete, before and after thermal aggression.

With the aim of reproducing the type of concrete exposure in a real fire, provoking real temperature differences between the different zones of a structural element, in the direct fire exposure test an average fire of a potential heat of 40 kg/m$^2$ was achieved, which is an average potential heat value which is produced in a fire inside a building. The heat value of dry wood is 19 MJ/kg and 40 kg of wood was used, generating heat energy of 181,640 kcal in the test [7].

In order to perform the fire testing, the test pieces were placed vertically to simulate the pillars of a building, their entire surface being in direct contact with the fire. All the test pieces were arranged on a steel grid placed on top of a test tray with a surface area of 1 m$^2$, loaded with 40 kg of chopped wood in order to enable greater oxygenation of the fire and encourage combustion at the source. The test lasted for 1 h, following the method standardized by ISO 834-11:2014 [49] and measuring the temperature in the upper, middle and lower sections of each test specimen every 5 min, using a compact infrared thermometer, "Testo 845"model. Once the hour-long fire exposure time had passed, the test specimens were slowly cooled down, dispersing their heat until reaching room temperature (20 °C). The fire test can be seen in Figure 2.

Before proceeding to perform the compression testing to breaking point, the test pieces were capped with sulfur mortar in accordance with regulation UNE-EN 12390-3:2009 [44]. In order to carry out the compressive strength testing, the test pieces were placed in the IBERTEST MIB-60/AM universal press and a preload of 10% of the maximum test load was applied, with the intention of ensuring that the upper plate was pressing uniformly on the upper surface of the test piece. Once the adjustment had been made, the compressive strength test was carried out using a piston in order to obtain the force and stroke data and then analyze the results which are relevant to the research objective, as shown in Figure 3.

In order to characterize the test pieces chemically and microscopically before and after the fire testing and after being subjected to compressive strength testing, the Scanning Electron Microscopy (SEM) technique was used. Using the X-ray detector (EDX), one-off and zonal microanalysis was carried out on the major elements in the samples showing natural breakage. The equipment used was the JEOL JSM-820 scanning electron microscope with microanalysis and the software was EDX Oxford ISIS-Link. In the study of the samples prepared using fresh concrete, CNF accumulation was

not detected due to the fact that perfect dispersion of the nanofiber addition was achieved, as such the test pieces also underwent thin-layer analysis, during which it was possible to observe the presence of nanofibers at 500 magnifications.

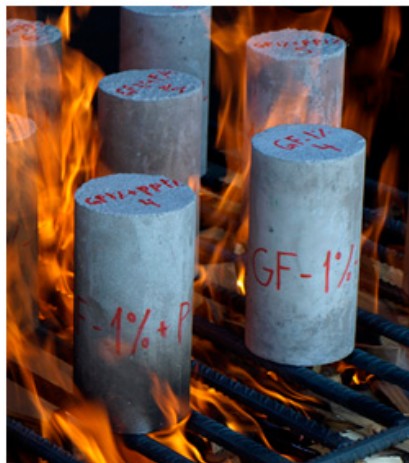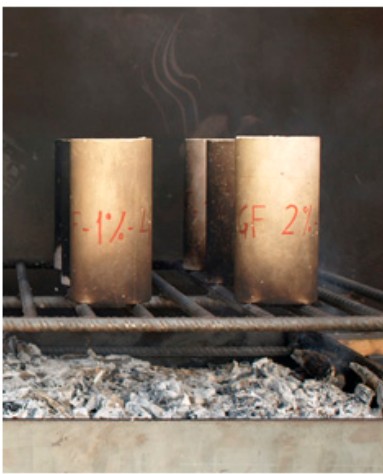

**Figure 2.** Fire test involving the entire surface of the test piece being directly exposed to the fire and taking measurements with an infrared thermometer.

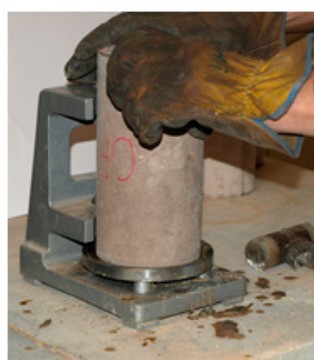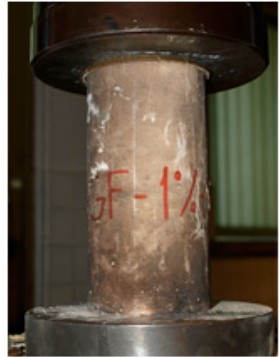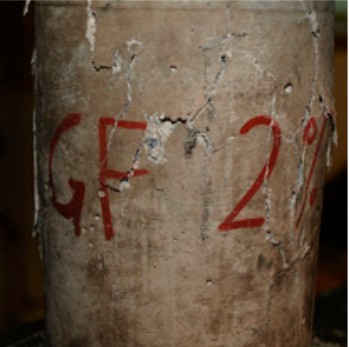

**Figure 3.** Capping of test pieces and compression testing to breaking point.

## 3. Results and Analysis

The results obtained in the compressive strength testing of the concrete before it underwent fire testing are shown in Figure 4, by means of the most representative graph for each type studied, reflecting the evolution of the strain in relation to tensile strength, in test pieces without addition and with percentages of 1% and 2% of CNFs. As can be appreciated, the strengths are markedly greater in the test pieces with the addition of CNFs, better behavior being achieved for percentages of 1% of nanofibers, where the strength achieved is notably greater as is the case with strain. However, the fact of incorporating a larger amount of CNFs does not improve strength in the test pieces.

Figure 5 shows us how the temperature evolves as the direct exposure to fire test progresses in the test pieces without addition and with the addition of 1% and 2% CNFs to the cement weight. As can be appreciated, the test provides us with significant data whereby the test pieces reach the highest temperature 10 min after starting the fire since it is a highly developed fire, reaching values of close to 600 °C in test pieces containing 2% of CNFs and values of 450 °C in test pieces without addition and with 1% of CNFs. The high temperature of the test pieces containing 2% of CNFs is due to the high heat transfer coefficient that this material possesses and that owing to its dispersion throughout the concrete mix the test piece increases in temperature evenly. Concrete containing 1% CNF in cement weight reaches similar temperature values to those reached in concrete without addition and 25%

lower than those reached in concrete with 2% CNFs in cement weight. In the case of concretes with 2% of CNFs, once the maximum temperature has been reached, there is a decrease in temperature faster than in concrete with 1% of CNFs, which translates into greater cracking of the concrete.

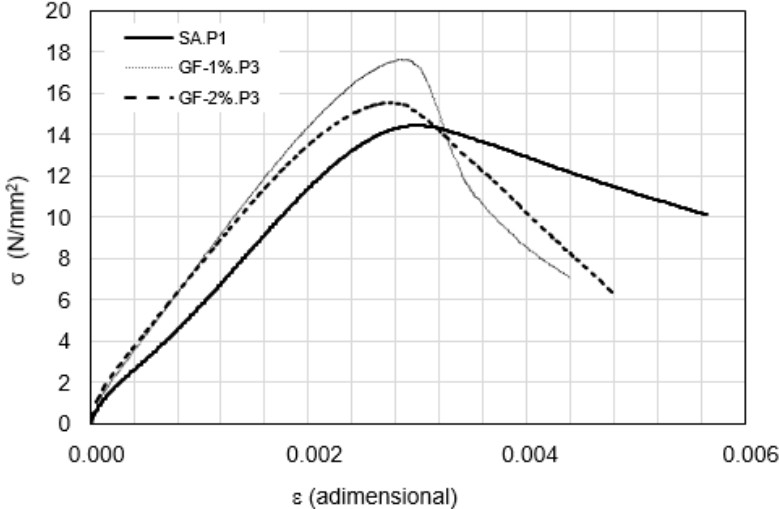

**Figure 4.** Evolution of strength (σ) in relation to strain (ε) in test pieces without addition, with percentages of 1% and 2% of CNFs.

The test pieces experience different temperature gradients depending on the areas where the flames reach them, as such, the heat transfer of the material can be observed, as well its absorption and dissipation capacity.

Focusing on test pieces made of traditional concrete (Figure 5a), without any kind of addition, we can see that the top of the 20 cm test piece displays temperature jumps, increasing 100 °C from the lower part where the fire is most direct, to the upper part, which reaches practically the same temperature as the middle section. This is due to the fact that concrete is able to dissipate the temperature on the surface of the material very well, and that in order to increase its internal temperature significant heat is required over a long time period.

Observing Figure 5b, it is worth highlighting that the temperature reached in the middle and upper areas of the test piece is practically the same, and in comparison with concretes without addition it has a lower thermal gradient. This is due to the 1% amount in cement weight of addition incorporated in the concrete, as a part of the calorific value of the fire is consumed in heating that proportion of CNFs dispersed in the cementitious mass, absorbing part of the energy and not transferring heat to the cement mass. At the lower part of the test piece, the temperature that is reached is greater, since the carbon nanofibers, due to their high heat transfer coefficient generate a temperature increase in this area of greater flame exposure. The concrete containing 1% CNFs in cement weight displays similar behavior in the cooling phase to the concrete without additions, where the temperature decrease is progressive and as such, the material becomes deformable and has less cracking.

Finally, Figure 5c is that which displays the highest temperature values in the entire test piece. It is due to the fact that doubling the amount of CNFs added provokes a greater temperature increase in the cement mass, since the dispersion of material with greater conductivity is high, transferring part of this heat to the cement matrix, and as such raising its temperature.

Figure 6 shows the evolution of strength in relation to strain, in test pieces without addition and with percentages of 1% and 2% of CNF addition in cement weight after undergoing fire testing. As can be observed, the higher strength levels are reached in test pieces containing 1% of CNFs in cement weight, since in the test pieces containing 2% of addition, strength levels are lower even in concretes without addition. Given the importance of how the strength-strain graph evolves in the behavior of the concrete, it is observed that in the concrete without addition, the graph is more homogeneous, depicting

a more ductile material. On the other hand, in the test pieces containing CNF additions, the gradient of the curves is more pronounced and imperfect, demonstrating that its behavior is more homogeneous. The most important piece of information is the strength increase in the test piece incorporating 1% of CNFs, its strength is greater compared to traditional concrete and features less strain, the material becomes tough when exposed to thermal aggression, but after reaching its maximum strength peak, it falls notably, losing ductility and becoming more fragile.

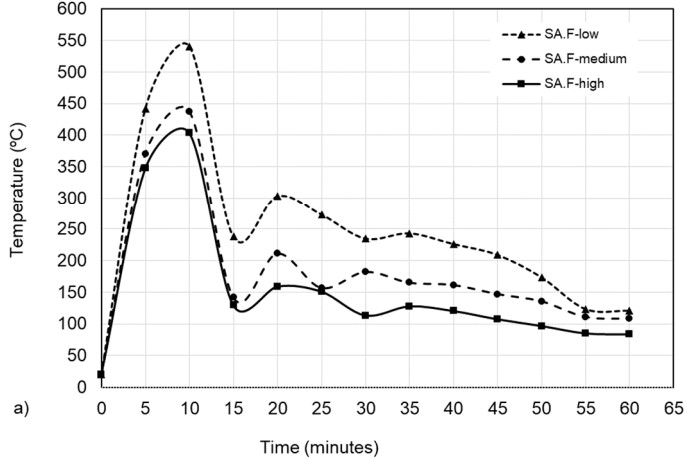

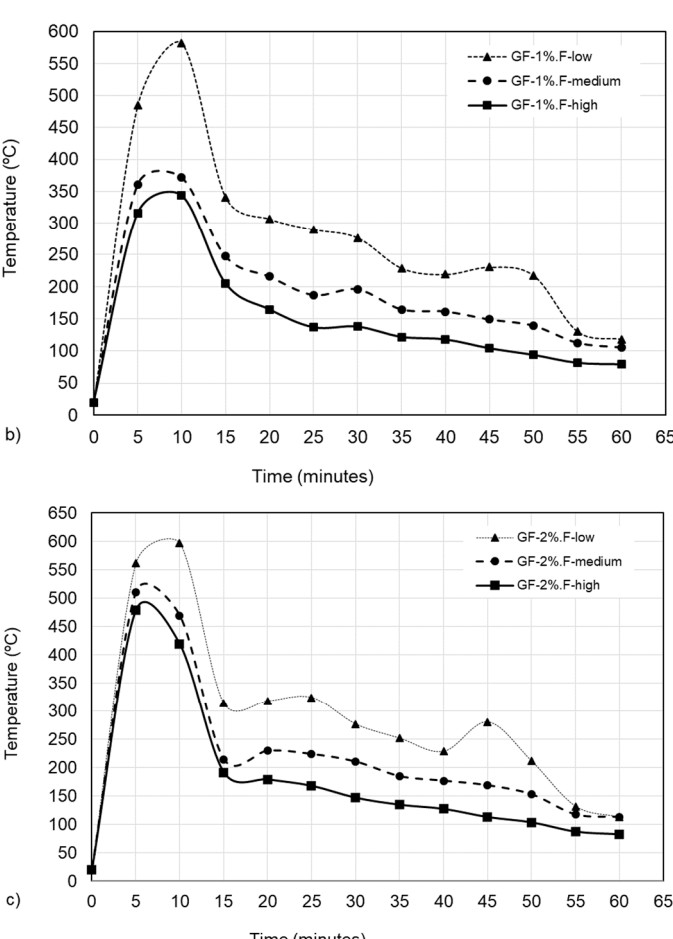

**Figure 5.** Evolution of temperature over time in test pieces without addition, with percentages of 1% and 2% of CNFs in their upper, middle and lower sections.

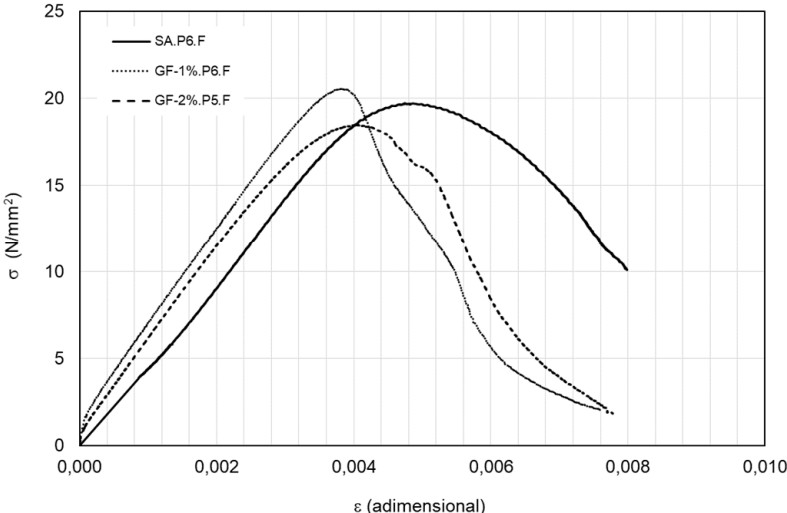

**Figure 6.** Evolution of strength ($\sigma$) in relation to strain ($\varepsilon$) in test pieces without addition, with percentages of 1% and 2% of polypropylene fibers and with percentages of 1% and 2% of steel fibers, after undergoing direct fire testing.

Table 4 shows the most representative mean data values from the strength-strain graphs, highlighting: maximum strength ($\sigma_{max}$), strain due to maximum strength ($\varepsilon_{max}$), ultimate strength values ($\sigma_u$), ultimate strain values ($\varepsilon_u$), maximum strain energy density ($E_{max}$) and ultimate strain energy density ($E_u$).

**Table 4.** Most representative mean values of the three specimens for each concrete type subjected to compressive strength testing and direct fire testing, with the different addition percentages.

|  | Addition Fibers | $\sigma_{max}$ (N/mm$^2$) | $\sigma_u$ (N/mm$^2$) | $\varepsilon_{max} \times 10^{-3}$ (Adimensional) | $\varepsilon_u \times 10^{-3}$ (Adimensional) | $E_{max} \times 10^{-2}$ (N/mm$^2$) | $E_u \times 10^{-2}$ (N/mm$^2$) |
|---|---|---|---|---|---|---|---|
| Compressive strength tests | Without addition | 14.372 | 10.063 | 3.117 | 5.956 | 2.755 | 6.217 |
|  | CNFs 1% | 17.183 | 6.877 | 3.020 | 4.690 | 3.161 | 5.340 |
|  | CNFs 2% | 15.943 | 6.382 | 3.013 | 5.420 | 3.063 | 5.781 |
| Fire + compressive strength test | Without addition | 19.556 | 10.938 | 4.864 | 8.466 | 5.406 | 11.199 |
|  | CNFs 1% | 22.187 | 2.221 | 4.068 | 7.981 | 5.393 | 9.590 |
|  | CNFs 2% | 18.630 | 1.86928 | 3.750 | 7.615 | 4.230 | 8.276 |

Figure 7 shows the signs and symptoms which occurred in compression testing to breaking point, in test pieces with and without additions and before and after direct exposure to fire. It is observed that the test pieces without addition display a more explosive break, with larger and deeper cracks, displaying a more deteriorated appearance of the material prior to structural collapse, however, the test pieces with the addition of CNFs, before and after being subjected to direct exposure to fire, display a less catastrophic type of breakage.

Figure 8 shows a comparison of the morphological characteristics at 300 magnifications of test pieces SA, GF-1% and GF-2% before and after the fire. In the test pieces which were not exposed to fire, excellent uniformity and bonding between the cementitious gel and the aggregate is observed and the formation of ettringite and portlandite is detected inside the existing pores.

In the test pieces exposed to fire, it can be observed that part of the ettringite and the clumps of $CaO \cdot SiO_2 \cdot H_2O$ (CSH gel) are destroyed and dispersed as a consequence of the heat, giving rise to a smoother textural appearance, where the acicular ettringite formations acquire rounded shapes and cracks appear due to heat expansion and contraction.

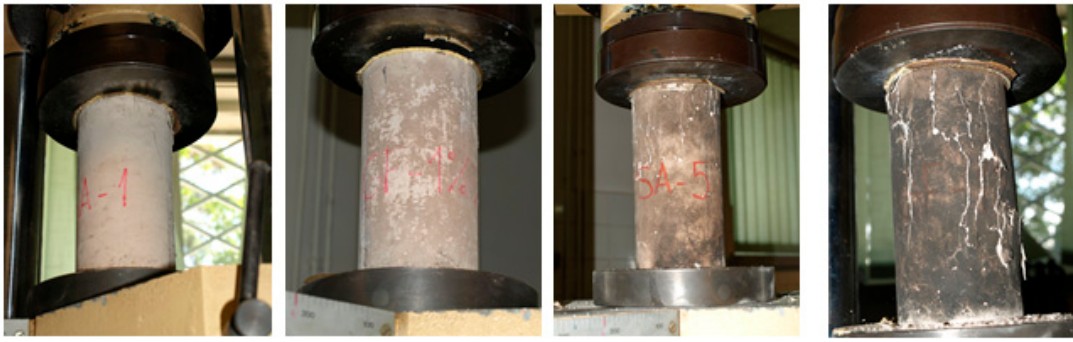

**Figure 7.** Compression testing to breaking point in test pieces without addition and with 1% CNFs, before and after fire testing.

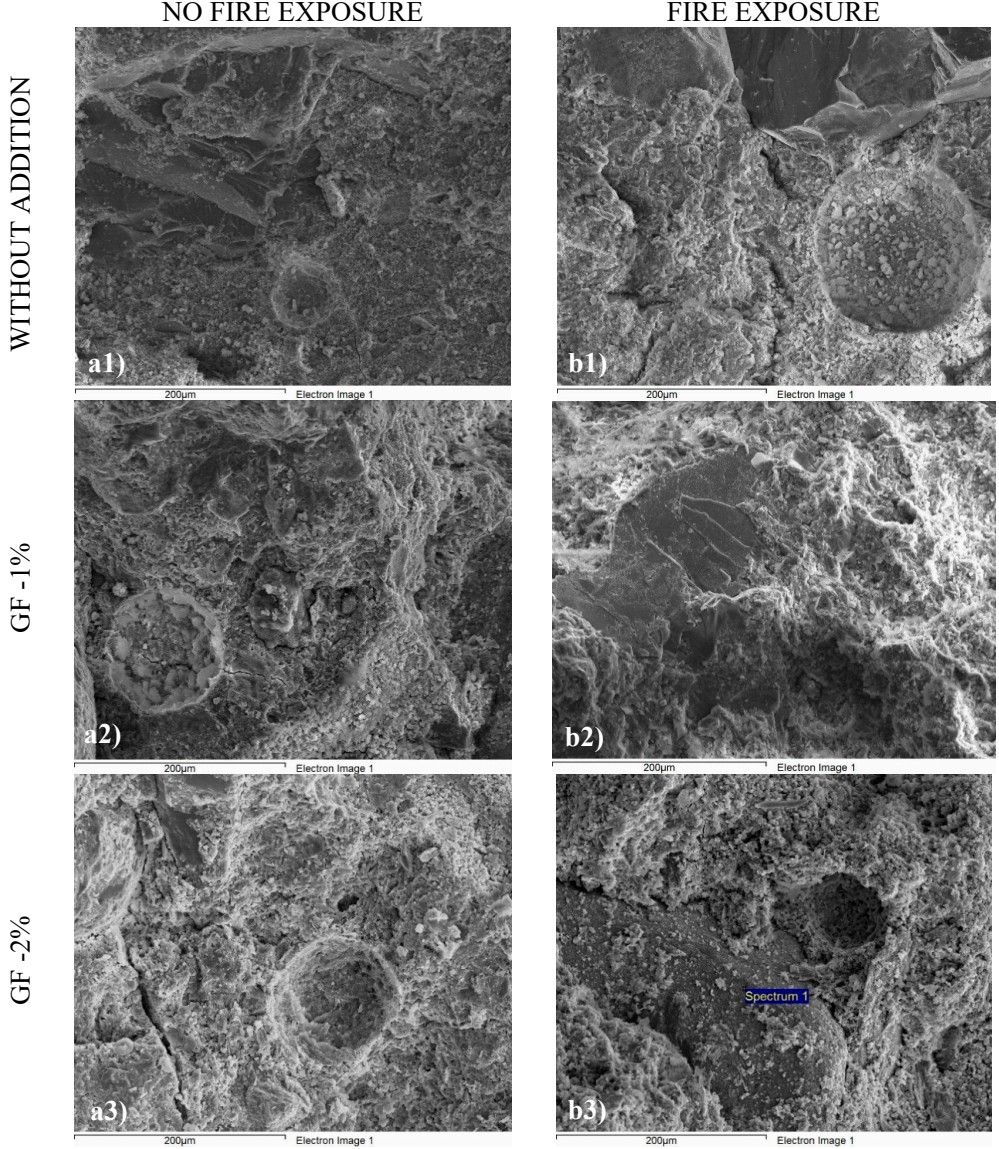

**Figure 8.** Scanning electron microscopy at 300 magnifications in test pieces without addition SA, and with the addition of CNFs in percentages of 1 and 2% by weight of cement GF-1%, GF-2%. (**a1**–**a3**) Before thermal aggression; (**b1**–**b3**) After thermal aggression.

In Figure 9, in the concrete without addition and not exposed to the fire, the presence of calcium silicate hydrate (C-S-H) gel can be observed, which is formed by hydrating cement, along with the

formation of portlandite and the acicular shapes which are typical of ettringite. In the concrete containing CNFs, it can be observed that the fibers point radially towards the exterior of the nodule, constituting a perfectly locked, dense crystalline mesh. Figure 9a2 shows the nested CSH nodules in a more advanced phase, where the silicate gel matrices are grouped together, leaving increasingly closed spaces, locked by the CSH fibers and the AFf phase. The calcium silicate hydrate is a principal cement compound and the main reason for its resistant properties, by means of its branches around the nodule and pointing towards the aggregate grains it bonds the compounds, anchoring the paste-aggregate mix and providing the cementitious matrix with cohesion and rigidity.

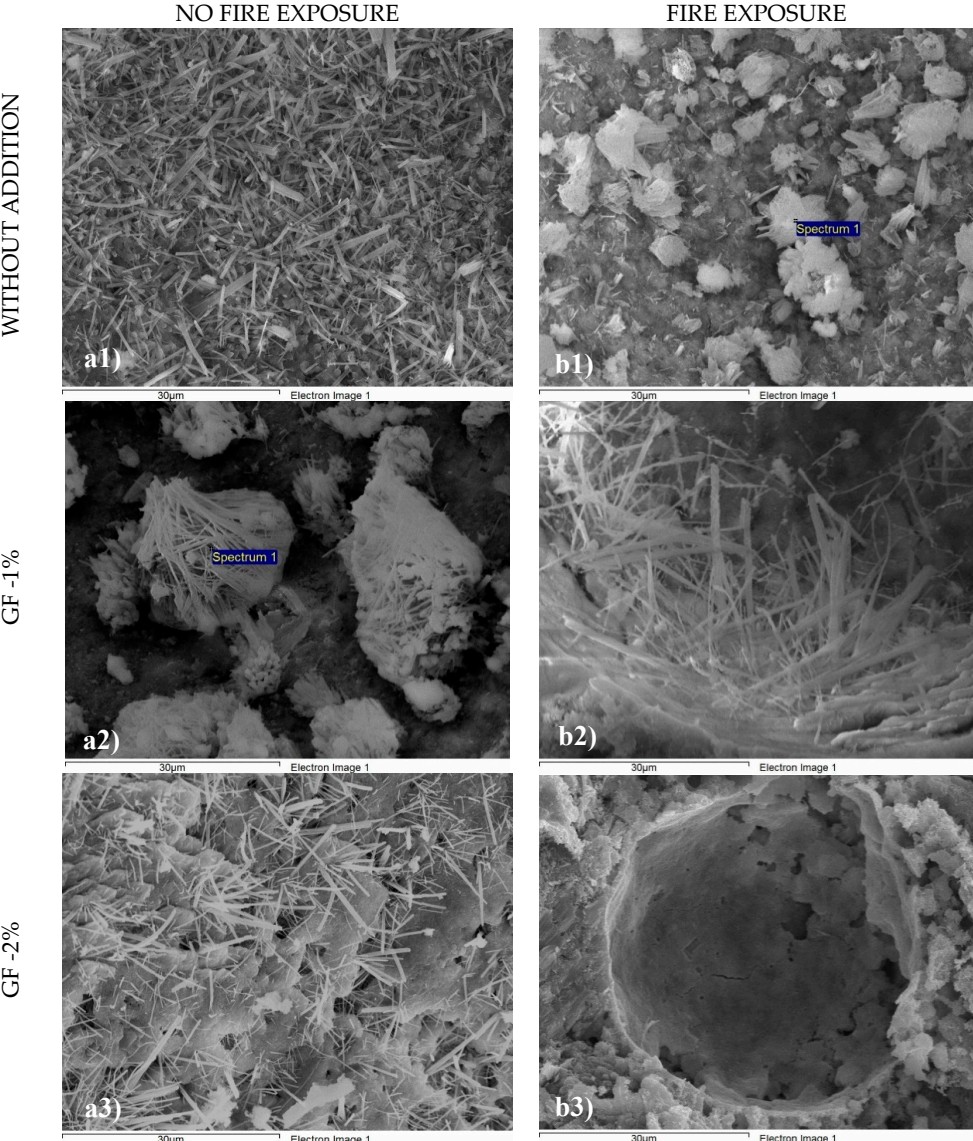

**Figure 9.** Scanning electron microscopy at 2000 magnifications in test pieces SA, GF-1% and GF-2%. (**a**) Before thermal aggression; (**b**) After thermal aggression.

After the fire, it can be observed that many of the CSH fibers in the concrete test pieces have been destroyed, thus reducing in number. However, in the concrete containing 1% of CNFs it is observed that the bonding of ettringite needles is not widespread, and the arrangement of the almost perfect crystalline mesh continues to exist, demonstrating the improved behavior in compressive strength testing of the test pieces containing CNFs. Finally, in the figure containing 2% of CNFs, we find a vacuole and inside the pore the ettringite fibers have disappeared due to thermal aggression, although

it has a good general appearance and its edges retain the CSH-fiber formation which confers resistance and cohesion to the structure.

Figure 10 making possible to observe that the figures related to the samples exposed to the fire (Figure 10b), possess a clearly common and characteristic feature, where all of the acicular and pointed shapes have become rounded. On the other hand, in the images of the concrete that was not exposed to the fire (Figure 10a), a clear, closed, globular structure can be observed with intermediate and advanced ettringite phases, with a healthy morphology and good integration of the paste with the calcium sulfate and aluminum.

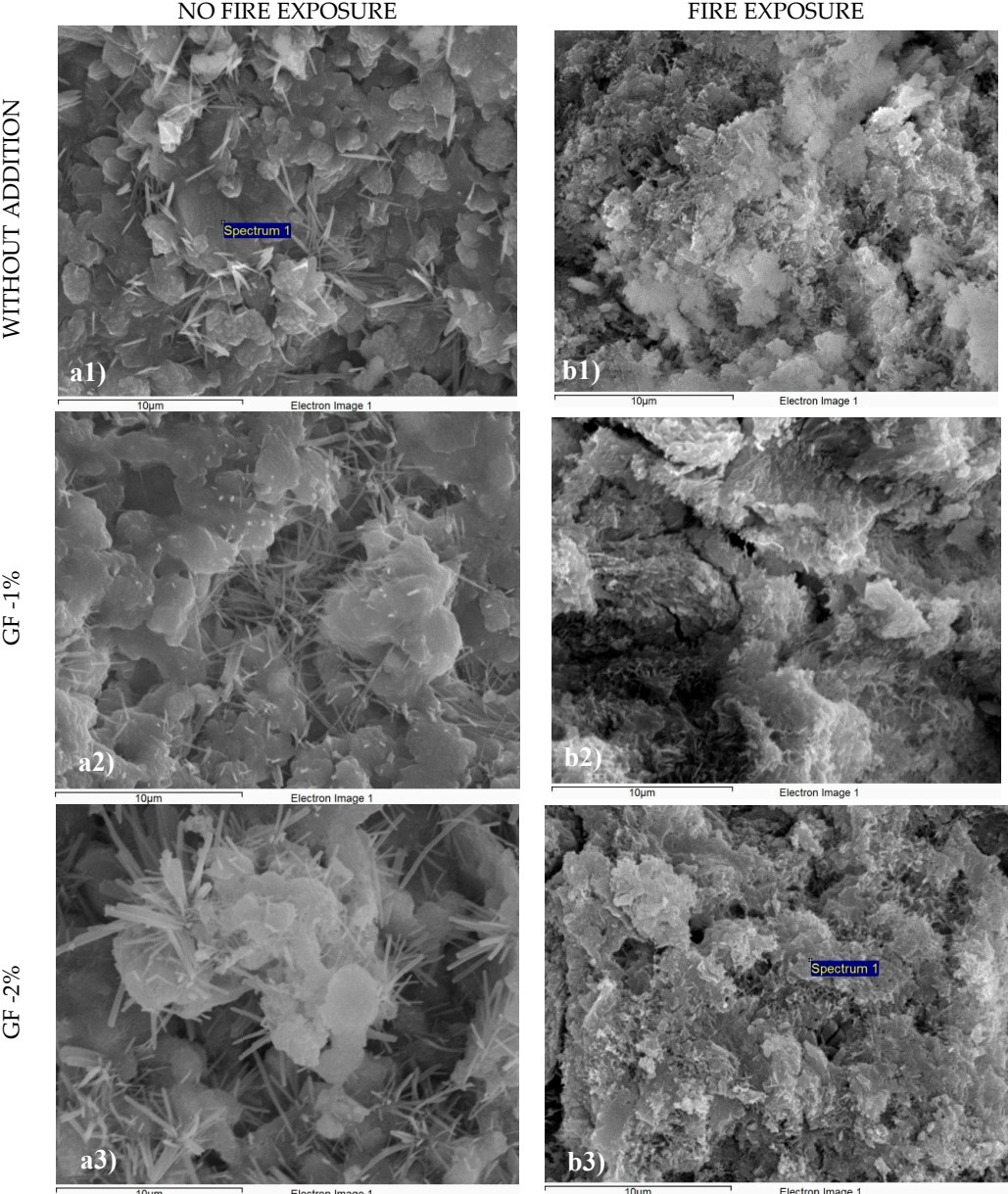

**Figure 10.** Scanning electron microscopy at 5000 magnifications in test pieces SA, GF-1% and GF-2%. (**a**) Before thermal aggression; (**b**) After thermal aggression.

Comparing Figure 10a1,b1, we can observe that in the left-hand image the aluminum and calcium silicates forming the ettringite have a highly-defined acicular shape and the parts featuring circular shapes are associated to the formation of portlandite. In Figure 10b1 the sulfates and carbonates react

with the action of the fire, melting much of the ettringite. The bubbles clearly disappear and the structure becomes more open.

In images 10b2,b3, despite the abrasion of the material, crystals can still be observed which withstand the thermal aggression of the fire, such that the join between the paste and the aggregates remains. The pores have practically disappeared and are occupied by the cementitious paste.

A study was also carried out concerning energy dispersive x-ray microanalysis, analyzing the elemental chemical composition based on a semi-quantitative analysis of concrete aggregates and the matrix of some of the samples of the groups studied. In Figures 11 and 12, the mineralogical descriptions are obtained from the spectrograms, where we observe the appearance of the presence of $SiO_2$ also known as CSH gel due to its formula: $CaO \cdot SiO_2 \cdot H_2O$, which is a calcium silicate that has undergone a hydration reaction and become a cementitious gel. We also find the presence of Na and Mg that it contains, originating from the rocks which are used to make the cement. The presence of gold (Au) and of titanium (Ti) is due to the coating which is incorporated in the samples in order to achieve a conductive surface and perform the mineralogical study of the test pieces.

Figure 11 shows the energy dispersive x-ray microanalysis at 100 magnifications of thin-layer test pieces incorporating the addition of 2% of CNFs to the cement weight, before and after the fire. If we focus on Figure 11a1, we can observe the presence of CNFs in the dark grey coloring distributed throughout the cementitious gel, which displays perfect dispersion in the mix. Figure 11a2 shows the spectrogram associated with the analysis taken at a point of image 11a1. In the analysis we can observe the presence of calcium, silica, sodium and magnesium which are typical in cement formation. Figure 11b2 shows the spectrogram of a point of the sample taken in image 11b1. We can see how the mineralogical analysis reveals the presence of a significant amount of iron (Fe), as a consequence of the fire exposure, which has decomposed the materials giving rise to the formation of iron and oxygen.

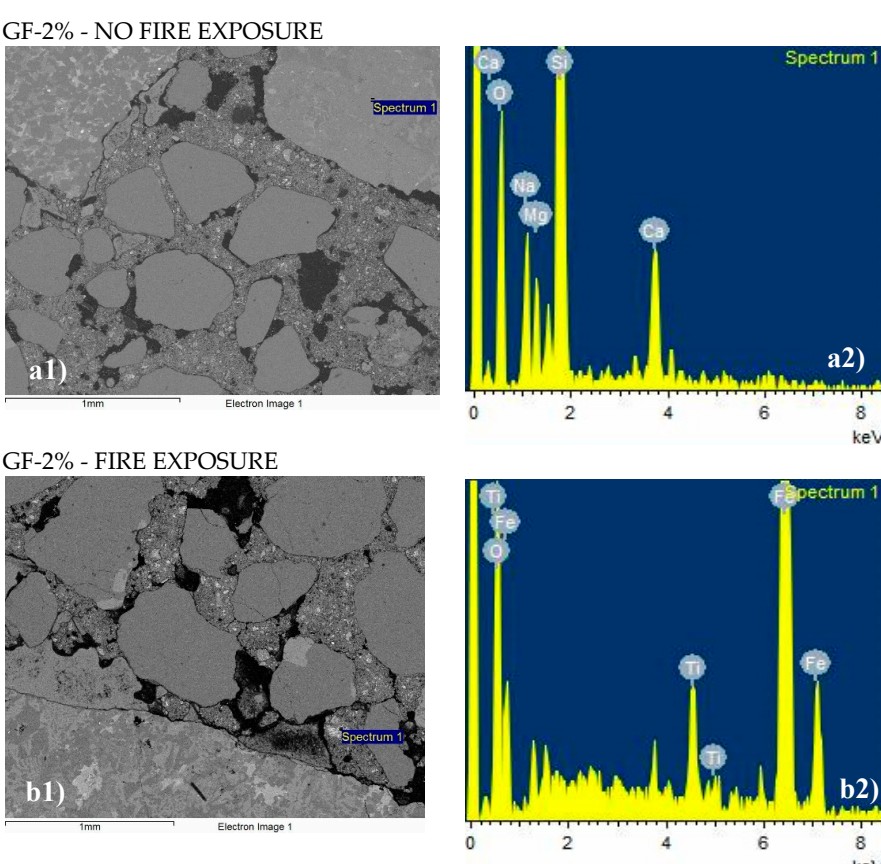

**Figure 11.** Energy dispersive x-ray microanalysis at 100 magnifications of thin-layer test pieces containing 2% of CNFs. (**a**) Before thermal aggression; (**b**) After thermal aggression.

Figure 12 below shows the energy dispersive x-ray microanalysis at 500 magnifications of thin-layer test pieces containing an addition of 2% of CNFs to the cement weight before and after the fire. The spectrogram that can be seen in image 12a2 shows the presence of carbon in high quantities, thereby demonstrating the existence of nanofibers. In image 12a1, we can see a large amount of dark grey dispersed pores which represent CNF accumulations. The white compounds relate to the aggregates present in the mixture, and the intermediate grey colors represent the existing cement gel. Image 12b2 shows the spectrogram of a point of the sample taken in image 12b1. We can see how the mineralogical analysis provides us with data concerning the presence of a significant amount of iron (Fe) and potassium (K), since as a consequence of the fire exposure the materials have decomposed giving rise to the formation of large amounts of iron, oxygen and potassium.

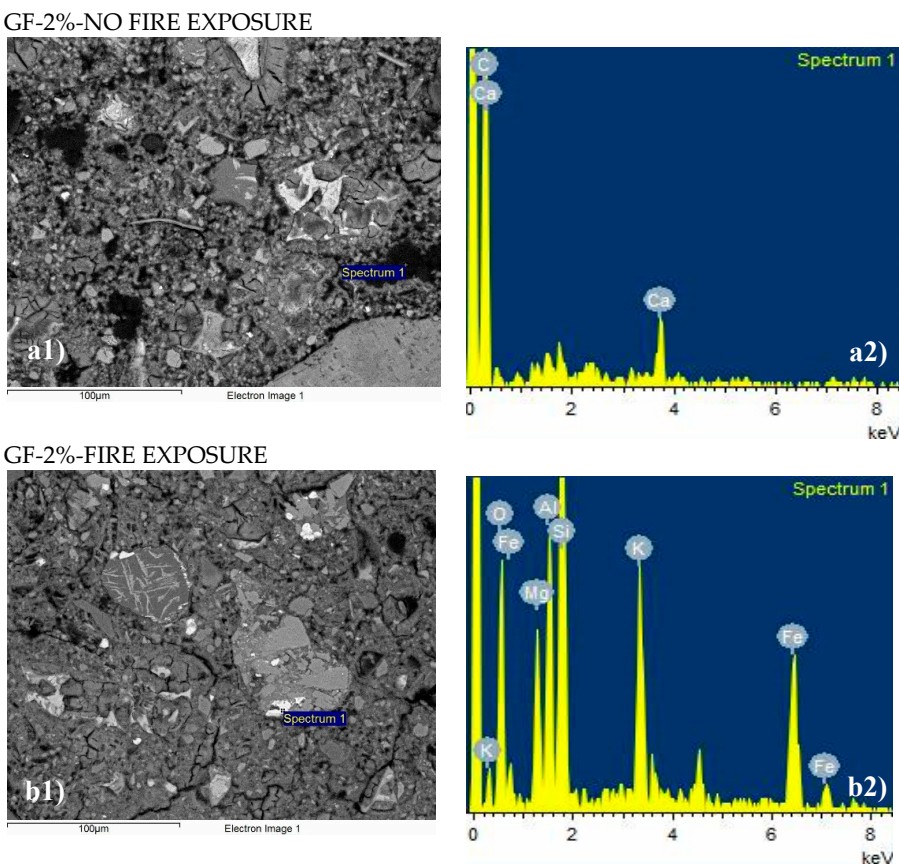

**Figure 12.** Energy dispersive x-ray microanalysis at 500 magnifications of thin-layer test pieces containing 2% of CNFs. (**a**) Before thermal aggression; (**b**) After thermal aggression.

## 4. Discussion

The analysis of the results of the compressive strength tests shown in Figure 13 make it possible to observe how the test pieces incorporating 1% or 2% of CNFs in cement weight increase their maximum strengths ($\sigma$max) compared to a traditional concrete without additions, and even when they have undergone direct fire testing, the strengths reached are much higher than 50% in the case of an addition amount of 1% of CNFs. What happens with maximum strains is worth special mention, since concretes without addition reach lower values when they are not exposed to the action of a fire but achieve strengths which are considerably higher, enabling us to deduce that the addition of CNFs in such percentages gives the concrete greater mechanical resistance, withstanding greater workloads and barely suffering material strain. The maximum strain energy density is closely related to the maximum strength reached and its maximum strain. The more energy a material absorbs, the better its mechanical behavior. As we can appreciate in Figure 13, the test specimens which best represent this parameter are

those containing 1% of CNFs. When we refer to concretes without addition and with a 2% addition of nanofibers, the strain energy density is very similar, enabling us to deduce than an increased addition to the concrete does not improve the material's behavior.

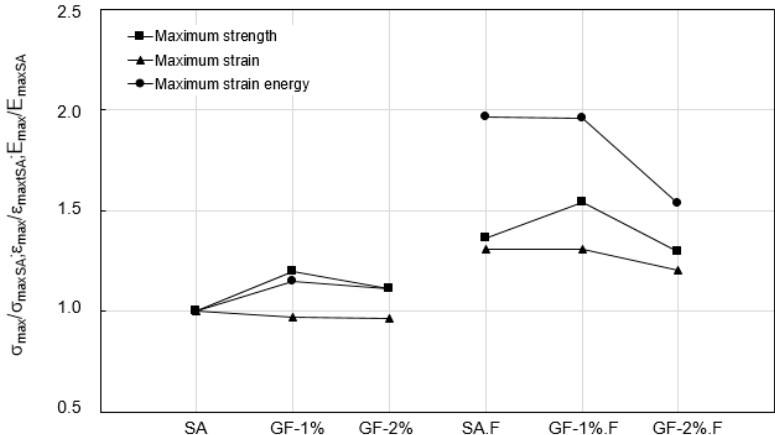

**Figure 13.** Evolution of the mean values associated with the maximum strengths and strains from compressive strength testing, in test pieces before and after direct fire testing.

Analyzing the ultimate strength values ($\sigma u$), Figure 14 enables us to compare the evolution of the mean values associated with the ultimate strengths and strains calculated based on compressive strength testing, before and after being subjected to direct fire exposure, using concrete without additions which has not been exposed to fire as a reference for the comparison. It is observed that the ultimate strength values are always lower in test pieces with the addition of CNFs, both before and after the fire, and that they are practically the same in concrete without addition following thermal aggression. When we observe the strain behavior for the ultimate strengths, we detect a unique phenomenon, which is that before being exposed to fire, the concretes with the addition of CNFs reach strain values which are also lower, however, in concretes with addition and which have been exposed to fire, despite their lower ultimate strengths, the strains achieved are much higher, with values approaching 25% in some cases. When we look at the data which the test concerning the ultimate strain energy density gives us, the behavior is similar to the ultimate strain, with lower values in concretes with addition and without being exposed to fire compared to traditional concrete. Once again, the values increase considerably when these concretes with addition are exposed to heat, reaching strain energy densities of almost 80% of the value of a traditional concrete.

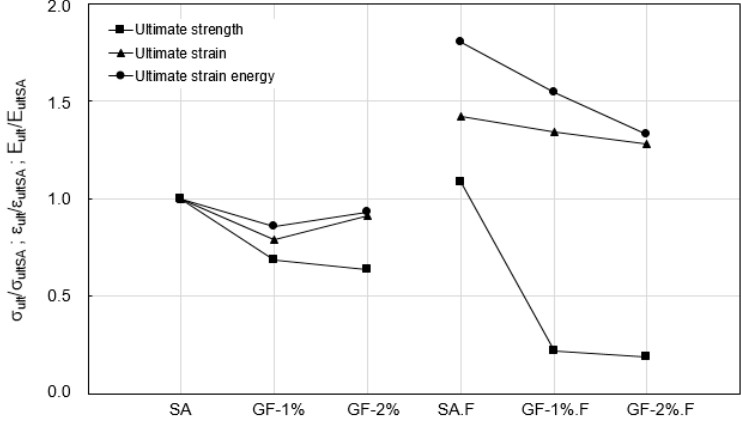

**Figure 14.** Evolution of the mean values associated with the ultimate strengths and strains from compressive strength testing, in test pieces before and after direct fire testing.

Figure 15 sets out the mean values of the temperatures reached by the test pieces with and without additions during direct fire testing. Generally, it can be observed that the line representing the most regular thermal behavior is that which incorporates a 1% addition of CNFs to the cement weight. The presence of excess CNFs, as in the group containing 2% of addition, causes the temperature to increase more rapidly than in the other groups studied due to its high heat transfer coefficient and significant proportion of nanofibers and as such the cooling phase is also slower, exposing the cement matrix to a higher temperature as a consequence of its thermal inertia. It is observed that the cooling down process is more gradual in the test pieces without addition and containing 1% of CNFs, particularly in the latter, which is an important piece of information concerning the behavior of the material with regard to the shrinkage that it could experience. The rapid temperature rise in the test pieces containing 2% of CNFs, encourages the appearance of cracks due to excessive expansion of the cementitious mass with the subsequent rapid drop in temperature in barely 50 min, this significant thermal fluctuation provokes enormous cracks in the material. The good thermal behavior of test pieces containing 1% of CNFs is surprising. This is the optimum addition amount, since it ensures that the cement mass absorbs less heat, this being absorbed by the CNFs, preventing the cement matrix from experiencing an excessive temperature increase. On the other hand, a gradual heat loss also results in less cracking of the material and consequently better mechanical behavior.

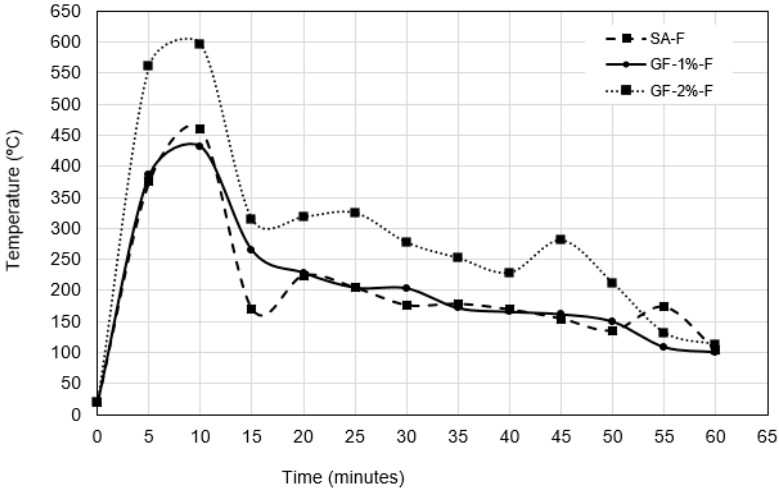

**Figure 15.** Evolution of the mean temperature reached in direct fire testing, in test pieces without additions and with 1% or 2% additions.

Table 5 compares the strengths between mass concrete test pieces without addition and with the addition of CNFs in different percentages, before and after being exposed to fire testing and the severity of the fire, understood to be the area enclosed in the temperature-time curves. As can be observed, the concretes with the addition of 1% of CNFs withstand strengths which are 20% greater than those tolerated by the test pieces without additions, whilst they increase by 10% when the percentage of CNFs is 2%. In the case where the concrete is exposed to direct fire, the concrete containing 1% of CNFs is that which performs the best, increasing its strength by 13.5% compared to the concrete without additions that is exposed to fire. On the other hand, when the addition is 2% of CNFs, strength is lost compared to the concrete without additions and exposed to fire, due to the fact that the severity of the fire tolerated is greater.

**Table 5.** Relationship between strengths, as a percentage, according to the type and percentage of addition and before and after fire testing and the severity of the fire.

| References | Compressive Strength Compared to No Addition (%) | | |
|---|---|---|---|
| | **SA** | **GF-1%** | **GF-2%** |
| SA | 100.00 | 119.56 | 110.93 |
| SA.F | 136.07 | 154.38 | 129.63 |
| Severity of the fire (°C min) | 12,600 | 12,855 | 14,295 |
| SA | 73.49 | 87.86 | 81.52 |
| SA.F | 100 | 113.46 | 95.27 |
| References | SA.F | GF-1%.F | GF-2%.F |
| | Compressive strength compared to SA.F (%) | | |

## 5. Conclusions

Analysis of the results obtained in the present research enable us to draw the following conclusions:

- Concrete with the addition of 1% of CNFs to the cement weight achieves a higher compressive strength of 19.5% than concrete without addition.
- Given the exposure to real fire, the incorporation of 2% of CNFs to the cement weight to the concrete, produces an increase in the average temperature reached by the concrete of the order of 25%.
- The addition to the concrete of CNFs in percentages of 1% by weight of cement, produces an improvement in the compressive strength of 13% with respect to a concrete without addition, when it is subjected to real fire.
- Scanning Electron Microscopy (SEM) shows a good paste-aggregate bond after real-fire testing in concrete with 1% of CNFs, which together with the severity of the fire reached and the type of cooling developed, decreases the effect "spalling".

**Author Contributions:** Conceptualization, R.S.S. and M.I.P.B.; Formal analysis, M.I.P.B. and M.d.l.N.G.G.; Investigation, K.J.H.M.; Methodology, R.S.S. and M.I.P.B.; Supervision, R.S.S. and M.I.P.B.; Writing—original draft, R.S.S. and M.I.P.B.; Writing—review & editing, M.d.l.N.G.G. and K.J.H.M. All authors have read and agreed to the published version of the manuscript.

**Funding:** This research received no external funding.

**Conflicts of Interest:** The authors declare no conflict of interest.

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
