# Peer review of "Analysis of the Behavior of Mass Concrete with the Addition of Carbon Nanofibers (CNFs) When Exposed to Fire"

_applsci, doi:10.3390/app10010117_

Round 1

Reviewer 1 Report

The paper is interesting but presents some deficiencies or lacks of the same since the research work carried out in my opinion is incomplete.

This are my questions :

- Why have you used CEM II instead of CEM I? The CEM II contains slag addition and using additions could mask the results. I think that if you use this type of cement you should make the comparison with CEM I

- How is the water cement ratio you have used?

- Have you taken into account the amount of water that the addition can absorb? When the same is done, the cement water ratio varies, since this material can absorb part of the water by varying the proportions of them.

- What is the granulometric curve of the aggregates that you have used?

- And the carbon nanoparticles?

- How much water do they absorb?

Figure 4 is missing characters (line 231)

- Have you taken into account what is the resistance of the material without subjecting it to fire? What resistant capacity does it reach?

-According to your paper, the temperature gradient is carried out according to UNE 1365 in which it says verbatim "thermocouples must be used". It does not specify how the measurements are made. If they are all at once, using 3 thermometers only one at different points with what can affect this measure.

- Other tests that should have been done would be flexion,

How can fire affect you?

Reviewer 2 Report

Both Abstract and  Conclusions should be modifying to clearing.

Reviewer 3 Report

Line 36 - reference?

Line 47 - an important disadvantage is the lack of binding between the nanoparticles and the cement mixture. The authors should mention this as well.

Line 62- can you explain what you mean by table dispersion

Table 3 - there is a FA in the table. It should be GF?

Line 150  and 153 - please add company name

Line 164 - what is meant by reach minimum level of resistance? 28 days is a standard test.

Line 181 - reference needed

Line 189 - company name needed

Figure 2 it is likely that the cylinders on the outer ring will get less heat than the other inside ones. consequently, the results are questionable. 

Please show in a sketch (top view) how the cylinders were placed inside the oven

Line 244 - “optimum addition”. how did the authors reach this conclusion? At 2% the hypothesis is that CNF high heat coefficient is responsible for the high temperature, yet at 1% the reasoning is that this is the optimum amount. How can such a conclusion be reached with just investigating two percentages? I would like the authors to deliberate on this issue.

Line 251 -so in the long run, if there is a fire, will the building blocks have to be changed if CNTs are incorporated? 

Line 273 - is this good or bad with respect to the health of the structure in the long run after a fire event?

Round 2

Reviewer 1 Report

The amount of water is an important aspect in the manufacture of concrete. The distribution of water has a significant influence on the resistant capacity of the material, if the same cement water ratio is added and maintained, it can have consequences both in durability and resistance. On the other hand, I consider that the curves of both aggregates and nanoparticles are tremendously important because of the amount of water they absorb.

Author Response

Comments and Suggestions for Authors REVIEWER 1

The amount of water is an important aspect in the manufacture of concrete. The distribution of water has a significant influence on the resistant capacity of the material, if the same cement water ratio is added and maintained, it can have consequences both in durability and resistance. On the other hand, I consider that the curves of both aggregates and nanoparticles are tremendously important because of the amount of water they absorb.

We appreciate and agree with the reviewer's observation, but in this work it was decided to compare the concretes maintaining the same w/c ratio in all the kneaded ones, so it will be considered for future investigations.

For the article, aggregate granulometry has been incorporated (Figure 1) and the following paragraph has been included in section 2. Materials and Methods:

The dosages which were used are shown in Table 2. The content of cement, sand, gravel and water is that used in each mix. The amount of addition used in each case is expressed in two ways: according to the percentage (%) of the cement weight and in kg/m3 of concrete. It was decided to use a constant w / c ratio regardless of the percentage of addition incorporated, being a determining factor in terms of strength and durability in the concrete. [43, 44]

Figure 1. Granulometric curves of aggregates. a) sand; b) gravel
